# Genetic Limitation and Conservation Implications in *Tetracentron sinense*: SNP-Based Analysis of Spatial Genetic Structure and Gene Flow

**DOI:** 10.3390/biology14091214

**Published:** 2025-09-08

**Authors:** Xiaojuan Liu, Xue Wang, Hongyan Han, Ting Pan, Mengxing Jia, Xiaohong Gan

**Affiliations:** 1Key Laboratory of Southwest Wildlife Resources Conservation, Ministry of Education, Nanchong 637009, China; liu1441219002@163.com (X.L.); wang166128@163.com (X.W.); hanhongyan@cwnu.edu.cn (H.H.); pantin_cwnu@163.com (T.P.); jmxcwnu@163.com (M.J.); 2College of Life Sciences, China West Normal University, Nanchong 637009, China

**Keywords:** fine-scale SGS, gene flow, protection strategy, SNP markers, *Tetracentron sinense* Oliv

## Abstract

This study investigated the genetic diversity and gene flow patterns of the endangered plant *Tetracentron sinense* in China to guide conservation efforts. The researchers analyzed four natural populations, revealing low genetic diversity and high inbreeding due to restricted seed and pollen dispersal, exacerbated by habitat fragmentation. Results indicated that gene flow is limited, with seeds and pollen typically traveling short distances (under 50 m in fragmented areas), leading to clustered genetic structures. The study underscores the urgent need for conservation strategies, such as habitat restoration and assisted gene flow, to prevent further genetic decline. These findings are crucial for protecting biodiversity and ensuring ecosystem stability.

## 1. Introduction

Spatial genetic structure (SGS), defined as the non-random distribution of genetic variation within populations, critically shapes evolutionary trajectories and conservation strategies for threatened species [1]. While broad-scale SGS analyses reveal regional connectivity patterns, their resolution remains insufficient to resolve localized gene flow dynamics or kinship clustering—key determinants of population resilience in stochastic environments. Fine-scale SGS quantification, in contrast, provides critical insights into intra-population genetic organization, serving as a proxy for short-term adaptive capacity and demographic buffering [2,3]. Persistent methodological constraints, however—including reliance on low-resolution markers (e.g., microsatellites) and single-population sampling frameworks [4]—hinder systematic evaluation of SGS heterogeneity across fragmented habitats. This knowledge gap impedes predictions of how dispersal limitation and anthropogenic disturbance interact to erode genetic resilience in endangered taxa.

*Tetracentron sinense* Oliv. is a large deciduous tree belonging to the genus *Tetracentron* (family Trochodendraceae) [5]. As a Tertiary relict species, it was once widely distributed across Europe, North America, and East Asia; however, due to climate change and human disturbance, it is now restricted to the alpine gorges of southwestern China, where it plays a crucial role in sustaining the stability of local ecosystems [6,7]. Notably, its xylem lacks vessels—a trait that endows it with significant research value for investigating the origin and phylogenetic evolution of angiosperms [8]. Additionally, *T. sinense* holds substantial utility in medicine, timber production, and horticulture. Regrettably, this high utility value has driven overlogging and overexploitation, leading to a continuous decline in its wild populations [9,10]. Currently, the species is listed in China’s National Key Protected Wild Plants (Second Class) and is also included in CITES Appendix III. While life-history studies of *T. sinense* have advanced, chronically low genetic diversity across populations [11,12] highlights the presence of unresolved gene flow bottlenecks. Recent microsatellite-based studies identified restricted dispersal as a driver of pronounced SGS in isolated populations [4]; yet, the limited number of loci and potential homoplasy in Simple Sequence Repeat (SSR) markers may limit the ability to resolve fine-scale genetic patterns [13,14]. Furthermore, whether these patterns are generalizable across the species’ ecogeographically heterogeneous distribution range remains untested—a critical uncertainty for developing comprehensive conservation strategies. Unlike SSRs, which are limited to a small set of potentially homoplasious loci, single nucleotide polymorphism (SNP) markers provide thousands of geographically distributed, biallelic, and codominant reference points. This substantially reduces variance in relatedness estimates and enables the detection of very recent gene flow events and fine-scale genetic patterns [15,16,17]. To date, no studies have investigated the SGS and gene flow patterns of *T. sinense* using SNP markers.

To address these limitations, we employed genome-wide single nucleotide polymorphism (SNP) markers generated via double digest restriction site-associated DNA sequencing (ddRAD−seq). We propose three non-mutually exclusive hypotheses: (1) Populations in fragmented habitats exhibit more pronounced dispersal limitations than those in contiguous forests, resulting in stronger spatial genetic clustering. (2) Kinship clustering is a direct consequence of restricted dispersal caused by habitat fragmentation. (3) Anthropogenic fragmentation enhances kinship clustering by compressing dispersal kernels. To test these hypotheses, we selected four natural *T. sinense* populations (comprising 378 individuals) based on phylogeographic significance, a habitat gradient spanning contiguous to fragmented areas, and ecological heterogeneity [6]. Employing approaches including landscape genomics and spatial autocorrelation analysis, we conducted an in-depth investigation into the fine-scale SGS patterns of *T. sinense* and the dynamics of its gene flow. The specific objectives of this study are as follows: (1) To investigate the fine-scale SGS patterns and underlying gene flow mechanisms in natural *T. sinense* populations. (2) To elucidate the fundamental drivers of low genetic diversity within *T. sinense* populations. (3) To provide a scientific basis for the conservation and management of *T. sinense*.

Our findings elucidate eco-evolutionary mechanisms constraining gene flow in relict trees while providing actionable guidelines for designing in situ conservation corridors and optimizing ex situ germplasm sampling strategies. This framework advances SGS theory while addressing urgent conservation needs for *T. sinense* and analogous taxa in fragmented ecosystems.

## 2. Materials and Methods

### 2.1. Study Design and Population Selection

To investigate spatial genetic structure (SGS) and contemporary gene flow in natural populations of *Tetracentron sinense*, we selected four representative populations spanning three evolutionary significant units (ESUs) identified by Li et al. [6]. Sampling strategies integrated phylogeographic history and latitudinal gradients: (1) the Baima Snow Mountain population (BMXS; 99°21′ E, 27°39′ N), representing a paleo-relict group on the western margin of the Hengduan Mountains; (2) the Meigu Dafengding population (MGFD; 103°08′ E, 28°46′ N), a glacial refugium during the Last Glacial Maximum (LGM); (3) the Leigong Mountain population (GLGS; 108°12′ E, 26°22′ N), added to assess latitudinal effects as the southernmost population in Guizhou Province; and (4) the Foping population (SXFP; 107°48′ E, 33°38′ N), reflecting post-Miocene northeastward expansion into the Qinling−Wuling Mountain corridor [6,18].

### 2.2. Field Investigation and Sample Collection

Prior to sampling, we systematically surveyed all natural *T. sinense* populations within the study area. To ensure experimental validity, three independent sampling patches were established per population based on two key criteria: (1) each patch must encompass a representative age structure of *T. sinense*, and (2) a minimum inter-patch distance of 1 km was maintained to avoid spatial overlap among individuals. Following rigorous evaluation of demographic composition and individual plant vigor, 12 optimal patches were selected for SGS analysis (Figure 1, Appendix A).

Within these patches, all *T. sinense* individuals underwent detailed biometric measurements, including GPS coordinates (longitude/latitude), diameter at breast height (DBH), vertical height, and phenotypic vitality. Following the methodologies of Rebertus et al. [19] and Zhang et al. [20], all individuals were classified into three age-classes based on DBH: (1) saplings (DBH < 7.5 cm), (2) adult trees (7.5 ≤ DBH < 22.5 cm), and (3) mature trees (DBH ≥ 22.5 cm). A total of 378 individuals were surveyed across the 12 patches. From each specimen, 6–10 intact young leaves were collected and immediately preserved in airtight plastic bags containing color-indicator silica gel for rapid desiccation. All samples were subsequently transported to the laboratory and stored at −80 °C for long-term preservation.

### 2.3. DNA Extraction and ddRAD Library Construction

Genomic DNA was extracted using the modified CTAB method and sent to Personal Biotechnology Co., Ltd., Shanghai, China [21]. The whole−genome DNA was completely digested with restriction enzymes (HindIII_MspI), and fragments of a specific length were recovered. The library was constructed following the double digest restriction site−associated DNA sequencing (dd−RAD) method. Utilizing second-generation sequencing technology (Next-Generation Sequencing, NGS) on the Illumina NovaSeq platform, the library was subjected to paired-end (PE) sequencing.

### 2.4. Obtaining Single Nucleotide Polymorphisms (SNPs)

After filtering the raw data using the sliding window method with fastp (v0.20.0), the high-quality filtered data was aligned to the reference genome (https://www.ncbi.nlm.nih.gov/genome/?term=tetracentron%20sinense, accessed on 4 September 2025) using the bwamem (0.7.12-r1039) program, with all alignment parameters set to default. Since duplicates generated during sequencing cannot serve as evidence for variant detection, the “MarkDuplicates” tool in the Picard software package Version 0.12.1 was used to remove duplicates. Additionally, reads near InDels are most prone to mapping errors. To minimize SNPs caused by mapping errors, due to resource constraints and version compatibility issues, we used the IndelRealigner command from an older version of the GATK program to realign all reads around InDels, thereby improving the accuracy of SNP calling [22].

The specific steps are as follows: (1) Alignments near InDels are generally unreliable and require realignment using known InDel information. The first step involves using the Realigner Target Creator command in the GATK software package version 3.5 to output a file containing all possible InDels; the second step uses the IndelRealigner command to realign all reads near InDels to improve the accuracy of SNP prediction. (2) The UnifiedGenotyper program was used to obtain SNP sites for the samples, with stand_call_conf set to 30 and stand_emit_conf set to 10 [23].

To ensure the reliability of the SNP sites, further filtering was applied to the obtained SNPs using the following criteria: (1) Fisher test of strand bias (FS) ≤ 60; (2) HaplotypeScore ≤ 13.0; (3) Mapping Quality (MQ) ≥ 40; (4) Quality Depth (QD) ≥ 2; (5) ReadPosRankSum ≥ −8.0; and (6) MQRankSum > −12.5.

### 2.5. Analysis of Population Genetic Diversity

Genetic diversity parameters, including expected heterozygosity (H_E_), observed heterozygosity (H_O_), and the inbreeding coefficient (Fis), were estimated for four *T. sinense* populations using the Stacks pipeline [24]. SNPs with minor allele frequency (MAF) > 0.05 were retained for downstream analyses to ensure robust estimates.

### 2.6. Fine-Scale Spatial Genetic Structure Analysis

SGS was analyzed at population, age-class, and patch levels using SPAGeDi 1.5 [25] and GenAlEx 6.503 [26]. The size of SGS, *Sp*, was quantified using the kinship coefficient Fij value in SPAGeDi 1.5 software [27], calculated by the following formula:*Sp* = −*bF*/(1 − F_(1)_)(1)
where F_(1)_ is the average kinship among individuals in the first distance class, and *bF* is the linear regression slope of kinship against the natural logarithm of the distance class [28]. The standard deviation and 95% confidence interval of Fij were estimated using 999 simulations. Fij = 0 indicates random kinship, Fij > 0 suggests higher relatedness than expected (positive SGS), and Fij < 0 implies lower relatedness (negative SGS) [27]. The multi-locus spatial autocorrelation coefficient r value was statistically analyzed using GenAlEx 6.503 software, with 999 simulations performed.

### 2.7. Gene Flow Analysis

#### 2.7.1. Direct Estimation of Gene Flow

The Cervus 3.0.7 software (http://www.fieldgenetics.com/pages/aboutCervus_Functions.jsp, accessed on 4 September 2025) was used for parentage analysis of each *T. sinense* patch [29]. During the analysis, saplings were treated as offspring, while mature and old trees were considered as candidate parents. Since the theoretical model of this method is suitable for cases where one or both parents are unknown, the parent pair with the highest and most significant Trio LOD value was automatically selected as the optimal parent combination. The parameters for the parentage analysis simulation were set as follows: the proportion of candidate parents was 0.9, the average genotyping error rate was 0.01, and the locus mismatch rate was 0.01 (referencing the default 80% confidence level). Subsequently, the GenAlEx 6.503 software was used to calculate the geographic distance matrix between individuals within each patch and to determine the reproductive distances of seed and pollen flow.

#### 2.7.2. Indirect Estimation of Gene Flow

Using the GenAlEx 6.503 software, the dispersal areas of pollen and seeds were estimated, and a random detection point along with 999 repeated sampling points were set to determine the significance of the spatial autocorrelation coefficient r [4]. r is a multi-locus dependency value used to test whether genetic variation at each locus changes with variations in neighboring loci. Its meaning is similar to that of the Moran’s I value. Ranging from −1 to 1, r indicates the degree of genetic similarity among individuals at different distance levels, thereby indirectly estimating the gene dispersal distance within each patch, i.e., the intersection of the autocorrelation coefficient’s curve with the *X*-axis. This is considered a geographically independent region, representing the effective diffusion area of gene flow [30].

Vekemans and Hardy [31] used a cubic equation to simulate the estimation of pollen and seed flow. The formula is as follows:(2)$$F_{\left(r\right)}=a+\mathrm{bln}\left(r\right)+c{\left[\ln\left(r\right)\right]}^{2}+d{\left[\ln\left(r\right)\right]}^{3}$$
where r is the geographic distance class, and F(*r*) is the corresponding correlation coefficient for that distance class, with r and F(*r*) values obtained from autocorrelation analysis. The R software 4.0.3 was employed to fit the parameters a, b, c, and d, and the formula was used to calculate the K value. The formula is as follows:K = 2c + 6ln(r1)(3)
where r1 is the midpoint of the first distance level.

## 3. Results

### 3.1. SNP Analysis

Following whole-genome sequencing of all samples and alignment with the *Tetracentron sinense* reference genome, we identified a total of 2,729,980 single nucleotide polymorphisms (SNPs) across all populations. Population-specific SNP counts were as follows: BMXS (97 samples, mean SNPs = 94,752), MGFD (129 samples, mean SNPs = 136,297), GLGS (105 samples, mean SNPs = 116,989), and SXFP (47 samples, mean SNPs = 161,766). The density distribution of SNPs across *T. sinense* chromosomes (CM026551.1–CM026574.1, corresponding to chromosomes 1–24; 2n = 48) was consistent with chromosome size variations, with SNP counts calculated in 0.5 Mb genomic windows. Notably, the chromosomal distribution pattern of these SNPs in *T. sinense* exhibited strong correlation with chromosome size gradients (r = 0.92, *p* < 0.001), suggesting proportional dispersion of genetic variation across the genome (Figure 2).

### 3.2. Analysis of Population Genetic Diversity

The four studied populations—BMXS, GLGS, MGFD, and SXFP—exhibited average observed heterozygosity (H_O_) values of 0.019, 0.019, 0.021, and 0.022, respectively, while their expected heterozygosity (H_E_) values were 0.083, 0.110, 0.130, and 0.079 (Table 1). Notably, H_O_ was consistently lower than H_E_ across all populations (H_O_ < H_E_), and these differences were statistically significant in all instances—suggesting the potential for inbreeding or selfing (Table 2). The inbreeding coefficient (Fis) further supported this observation, with values ranging from 0.147 (SXFP) to 0.304 (MGFD), the latter being significantly higher than those of the other populations.

### 3.3. FSGS Analysis at the Population Level

The fine-scale spatial genetic structure (FSGS) analysis revealed population-level consistencies and divergences in spatial genetic organization across *T*. *sinense* populations (Figure 3, Table 3). All populations exhibited a positive SGS at short distances (≤86–210 m), with kinship coefficients (Fij) transitioning to neutral or negative values at extended spatial scales, indicating limited gene dispersal and localized kinship clustering.

The BMXS population demonstrated the strongest geographical isolation, characterized by a steep regression slope (*bF* = −0.0204), elevated initial kinship (F_(1)_ = 0.0531), and high SGS intensity (*Sp* = 0.021), with a significant positive SGS persisting up to 470 m before shifting to negative correlations beyond 495 m. In contrast, the MGFD population displayed attenuated spatial structuring, evidenced by shallower slope parameters (*bF* = −0.0075), lower F_(1)_ (0.0176), and reduced *Sp* (0.0076), with positive SGS confined to ≤150 m followed by neutral (151–325 m) and negative (325–670 m) genetic correlations. Intermediate isolation patterns emerged in the SXFP population, featuring alternating SGS regimes with positive correlations at ≤99 m and 200–260 m interspersed with neutral (100–130 m, 160–200 m) and negative (130–160 m, 375–700 m) intervals, reflected in moderate metrics (*bF* = −0.0133; F_(1)_ = 0.0297; *Sp* = 0.013). Notably, the GLGS population exhibited the most spatially restricted genetic clustering, with positive SGS limited to ≤86 m and complete structural dissolution at larger distances, despite maintaining steep regression slopes (*bF* = −0.012) and high initial kinship (F_(1)_ = 0.0362).

### 3.4. FSGS Analysis at the Age-Class Level

The SGS of *T. sinense* populations exhibited age-class-dependent patterns across the studied sites, with notable similarities and variations in scale and intensity (Figure 4, Table 4). Saplings consistently demonstrated stronger SGS at shorter spatial scales compared to adult and old trees, as evidenced by higher F_(1)_ and *Sp* values in the first distance class across all populations where saplings were present (BMXS, MGFD, GLGS). Specifically, significant SGS in saplings occurred within 12–115 m, with BMXS and GLGS populations showing broader spatial autocorrelation ranges (52–115 m) compared to MGFD (12–29 m). Adult trees generally displayed intermediate SGS intensity, with significant spatial clustering occurring at variable distances: 15–45 m in GLGS, 18–79 m in SXFP, and 71–73 m in MGFD, while BMXS exhibited a unique bimodal pattern (20–40 m). Old trees predominantly showed fragmented or extended SGS patterns, with significant autocorrelation detected both at very short scales (9–18 m in MGFD and BMXS) and over longer distances (35–70 m in BMXS, 35–37 m in MGFD, and >53 m in SXFP and GLGS).

Population−specific divergences emerged in SGS hierarchy among age-classes. BMXS and MGFD followed a sapling > old tree > adult tree gradient in SGS strength, whereas GLGS reversed this trend (old tree > adult tree > sapling) with significantly elevated *Sp* values in old trees. Notably, SXFP lacked saplings and revealed stronger SGS in adult trees compared to old trees, contrasting with other populations. Adult trees in MGFD uniquely exhibited negative F_(1)_ values within their primary spatial scale (71–73 m), indicating no significant genetic structure.

### 3.5. FSGS Analysis at the Patch Level

The SGS of *T. sinense* exhibited pronounced heterogeneity among patches within geographic populations (Figure 5, Table 5). In the BMXS population, significant positive SGS was confined to a single patch (YC) at fine spatial scales (10–28 m and 58–63 m; *Sp* = 0.0107), contrasting with unstructured neighboring patches (*Sp*: 0.0030–0.0032). Similarly, the MGFD population displayed intra-population divergence, with two patches (MB and MC) showing positive SGS (16–28 m; *Sp*: 0.0063–0.0119) and one neutral patch (*Sp* = 0.0006). Strikingly, the GLGS population demonstrated opposing SGS patterns: patch (GB) exhibited alternating positive (11 m, 23–26 m) and negative spatial correlations (33–37 m; *Sp* = 0.0177). In the SXFP population, only the FB patch displayed significant positive SGS (19–32 m and 50 m; *Sp* = 0.0177), with others showing weak structuring (*Sp*: 0.0020–0.0103).

### 3.6. Gene Flow Analysis

#### 3.6.1. Direct Estimation of Gene Flow

Analysis of direct gene flow estimation across four *T. sinense* populations revealed consistent inverse relationships between dispersal distances and genetic connectivity for both seeds and pollen, though population-specific variations in dispersal scales were evident (Figure 6, Table 6). All populations exhibited declining dispersal frequencies with increasing inter-individual distances, with seed dispersal generally spanning broader spatial scales than pollen dispersal. For instance, the BMXS population demonstrated markedly longer range dispersal, with seed distances averaging 160.62 m (max 463.52 m) compared to pollen averages of 110.35 m (max 386.20 m), while 75% of seeds and 86% of pollen remained within 170 m. Contrastingly, the MGFD and GLGS populations displayed constrained dispersal patterns, with seed averages of 32.44 m and 29.60 m (maxima 61.76 m and 75.62 m) and pollen averages of 30.59 m and 33.35 m (maxima 78.94 m and 84.06 m), respectively, where over 64% of seeds and 71–91% of pollen were retained within 40–60 m radii. The SXFP population occupied an intermediate position, with seed and pollen averages of 78.73 m and 70.64 m (maxima 187.69 m and 156.67 m) and equivalent 62% retention within 90 m ranges.

#### 3.6.2. Indirect Estimation of Gene Flow

The analysis of indirect gene flow estimation across four *T. sinense* populations revealed consistent patterns and inter-population divergences in SGS and dispersal dynamics (Figure 7, Table 7). Three populations (BMXS, MGFD, SXFP) showed coexistence of patches with spatially random genetic variation (YA, MA, FA/FC) and those exhibiting significant SGS at defined intervals: YB (20–40 m) and YC (55–65 m) in BMXS; MB (40 m) and MC (30 m) in MGFD; and FB (19–30 m and 46–53 m) in SXFP. In contrast, GLGS displayed uniform SGS across all patches (GA:18–27 m, GB:17 m, GC:28–32 m), indicating population-wide genetic clustering. Effective dispersal distances varied substantially between populations, with ranges of 16.14–29.42 m in BMXS (shortest), 42.66–46.81 m in MGFD (longest continuous), 37.03–66.12 m in SXFP (maximum range), and 23.00–31.00 m in GLGS (moderate uniformity). Notably, populations containing random−variation patches showed greater dispersal distance heterogeneity (Δ13.28–29.09 m) compared to GLGS (Δ8.00 m). A consistent inverse relationship emerged between SGS spatial extent and dispersal capacity—patches with longer SGS detection ranges (e.g., BMXS−YC:55–65 m) displayed shorter dispersal distances (29.42 m), while those lacking SGS (e.g., SXFP−FA) achieved maximal dispersal (66.12 m).

These patterns suggest microenvironment−driven dispersal constraints, where habitat fragmentation compresses dispersal kernels in structured patches while allowing occasional long-distance gene flow in spatially random counterparts.

## 4. Discussion

### 4.1. Genetic Diversity of Natural Populations of T. sinense

Genetic diversity analysis of four natural populations of *Tetracentron sinense* (BMXS, GLGS, MGFD, and SXFP) revealed a consistent pattern: observed heterozygosity (H_O_, 0.019–0.022) was consistently lower than expected heterozygosity (H_E_, 0.079–0.130) (Table 1), and significantly lower than reported values in other relict angiosperms—such as *silver fir* (H_O_ ≈ 0.21–0.33) and *Pterocarya macroptera* (H_O_ ≈ 0.063–0.097) [32,33]. This discrepancy underscores the inherently low genetic diversity characterizing natural populations of *T. sinense*.

Such low genetic diversity (mean H_E_ = 0.101) and high inbreeding levels (inbreeding coefficient Fis = 0.147–0.304) carry critical implications for the species’ long-term viability. First, reduced heterozygosity constrains adaptive potential, as exemplified by the 100% seedling mortality in the SXFP population—where inbreeding depression likely compromised stress resistance (e.g., drought and pathogen tolerance). Second, elevated Fis values indicate increased homozygosity of deleterious recessive alleles, further diminishing seedling fitness—a probable key driver of the species’ endangered status. Moreover, limited genetic variation impairs populations’ capacity to buffer against environmental perturbations (e.g., climate fluctuations), heightening the risk of local extinction. Intriguingly, SNP-based H_E_ values for *T. sinense* were markedly lower than those obtained using SSR markers in the same species (H_E_ = 0.598) [4] and related taxa, such as *Pteroceltis tatarinowii* (H_E_ = 0.426–0.444) and *Fagus hayatae* (H_E_ = 0.639) [34,35]. This divergence may stem from methodological differences: SNPs provide genome−wide coverage with higher resolution, enabling more accurate estimates of genetic diversity compared to SSR markers, which are prone to homoplasy and selective neutrality biases. Nevertheless, restriction site−associated DNA sequencing (RAD−seq)—a technique widely used for SNP analysis—has limitations: it may introduce biases toward non-coding regions and requires a high−quality reference genome to ensure optimal sequence alignment and variant calling, a constraint when compared to the more universally applicable SSR protocols [36].

### 4.2. Fine-Scale SGS of T. sinense

The multi-scale analysis of spatial genetic structure (SGS) in *T. sinense* populations unveils critical insights into the interplay between ecological constraints, life-history traits, and landscape heterogeneity in shaping fine−scale genetic organization. At the population level, all studied populations exhibited significant SGS (*Sp* = 0.0076–0.0210), characterized by kinship clustering at short distances (<210 m) and neutral or negative genetic correlations at broader scales. This pattern aligns with expectations for tree species experiencing restricted gene flow due to localized seed dispersal and habitat fragmentation [37,38]. The observed *Sp* values place *T. sinense* within an intermediate SGS range compared to other angiosperms—higher than *Acer saccharum* (*Sp* = 0.0102) and *Quercus robur* (*Sp* = 0.0080) but lower than *Cunninghamia lanceolata* (*Sp* = 0.0327) and *Ulmus chenmoui* (*Sp* = 0.0293) [1,31,39,40]. Notably, the BMXS population displayed the strongest SGS (*Sp* = 0.0210), likely driven by its high seedling density (DBH <3 cm) and pronounced geographical isolation, which amplify kinship clustering. In contrast, the reduced SGS intensity in MGFD (*Sp* = 0.0076) may reflect its expansive terrain and suboptimal grazing conditions, which disrupt seedling establishment while facilitating broader pollen-mediated gene flow. These population-level divergences underscore how microhabitat features and anthropogenic pressures modulate genetic structure even within a single species.

Age−class stratification further clarified the temporal dynamics underlying SGS formation. Compared to adult and old trees, saplings exhibited intensified SGS (e.g., BMXS population: F_(1)_ = 0.0531 within the 52–115 m distance class). This pattern supports the predominance of localized seed dispersal, which aligns with gravity−driven seed fall and limited seedling mobility. In contrast, the fragmented or extended SGS patterns observed in old trees (e.g., BMXS: 35–70 m; MGFD: 35–37 m) likely serve as archives of historical dispersal events or clonal propagation, reflecting long-term recruitment bottlenecks. Notably, two population−specific recruitment challenges were identified: a reversed SGS hierarchy (old tree > adult > sapling) in the GLGS population, and a complete absence of saplings in the SXFP population. The extremely high seedling mortality of *T. sinense* is likely driven by the interaction of multiple factors. Field surveys revealed that dense understory vegetation in the GLGS and SXFP populations intensifies competition for light and nutrients. In these habitats, *T. sinense* seedlings are outcompeted by both conspecifics and associated species, thereby contributing to their high mortality rates. Additionally, young trees are scarce in the MGFD population, with the MB patch lacking young individuals entirely. Characterized by open terrain, this site is frequently visited by cattle and sheep grazed by local herders, leading to extensive browsing on *T. sinense* seedlings—providing evidence that predation pressure also plays a role in driving seedling mortality. These observations indicate that seedling mortality is influenced not only by genetic factors (e.g., inbreeding depression) but also by ecological constraints. This highlights the need for more rigorous quantitative analyses to disentangle the relative contributions of these effects. Furthermore, such age-class disparities emphasize the importance of longitudinal studies to distinguish between transient and persistent genetic patterns in long-lived species.

Heterogeneity in SGS across different patches within the same population (e.g., YC versus YA/YB in BMXS) may stem from microhabitat variations (e.g., slope steepness), as well as human disturbances and historical factors. For example, the YA and YB patches—distributed as narrow, elongated belts—exhibit weaker SGS than the YC patch, which is situated in a mountain gully with a relatively steep slope. Among the three patches in GLGS, the GB patch is separated by mountain roads, which restrict gene flow to a certain extent. Moreover, the clustered distribution of *T. sinense* in this area results in short-distance seed dispersal, causing GB to display stronger SGS than the GA and GC patches. In the FB patch, the clustered distribution of old trees accounts for its distinct SGS profile. These diverse factors collectively indicate that contemporary dispersal limitations and legacy effects jointly contribute to genetic clustering. In contrast, neutral or negative SGS in spatially fragmented patches (e.g., GLGS−GC: *Sp* = 0.0066 at 57–62 m) may reflect pollen swamping or stochastic mortality events that disrupt kinship structures. Notably, the lower *Sp* values observed in this study (e.g., GLGS: *Sp* = 0.012) compared to SSR−based studies (*Sp* = 0.0305) suggest potential methodological influences, including differences in marker resolution and sampling design.

### 4.3. Gene Flow of Natural Population of T. sinense

The integration of direct and indirect gene flow estimations in *T. sinense* populations reveals critical insights into the spatial dynamics and ecological constraints shaping genetic connectivity in this species. Our findings demonstrate pronounced heterogeneity in dispersal patterns across populations, driven by microhabitat heterogeneity and species−specific biological traits.

Direct estimation revealed a consistent inverse relationship between dispersal distances and genetic connectivity for both seeds and pollen, albeit with marked population−level variations (Figure 6, Table 6). Notably, the BMXS population exhibited broader dispersal scales (seed: 160.62 m average, pollen: 110.35 m), contrasting sharply with the truncated dispersal in MGFD and GLGS populations (seed: <33 m, pollen: <33 m). This disparity likely reflects habitat continuity in BMXS and SXFP (intermediate dispersal: 80–90 m), which facilitates long−distance dispersal vectors, versus fragmented microhabitats in MGFD and GLGS that impose physical or behavioral constraints on pollinators and seed dispersers. The stronger distance-dependent decay in pollen flow compared to seed dispersal, particularly evident in BMXS (18% greater restriction within 170 m), underscores differential efficiency between biotic pollinators (e.g., Xylocopa dissimilis) and abiotic seed dispersal mechanisms [41]. These patterns align with observations in other insect−pollinated species, where fragmented habitats amplify pollen limitation due to reduced pollinator mobility [31,42].

Indirect estimation via SGS analysis further corroborated microenvironmental modulation of gene flow (Figure 7, Table 7). Populations with mixed genetic patches (BMXS, MGFD, SXFP) displayed greater dispersal heterogeneity (Δ13.28–29.09 m) compared to GLGS (Δ8.00 m), which exhibited uniform SGS clustering. The inverse relationship between SGS spatial extent and dispersal capacity—e.g., longer SGS detection ranges (BMXS-YC: 55–65 m) coinciding with shorter dispersal distances (29.42 m)—suggests habitat fragmentation compresses dispersal kernels in structured patches while allowing sporadic long−distance gene flow in spatially random areas. This mirrors findings in fragmented forest species, where limited dispersal reinforces genetic clustering [43].

The above results all indicate that gene flow in *T. sinense* is constrained by contemporary landscape features and its unique biological and ecological traits: (1) As a species exhibiting both anemophily and entomophily, with self−compatibility and cross−fertility, *T. sinense* theoretically benefits from multiple pollination vectors. However, the overlap between its flowering period (June–July) and monsoon rains reduces pollinator activity, a phenomenon also documented in other pollinator−dependent species [8]. (2) Its seeds are embedded in fruiting spikes that partially detach upon maturation, significantly limiting seed dispersal efficiency. This results in a markedly lower dispersal capacity compared to wind−adapted species like *Handeliodendron bodinieri* [44]. (3) Mountainous topography fragments populations into linear distributions, isolating subpopulations and impeding pollen and seed movement [4,45]; the mountain roads in the GLGS−GB patch have reduced the gene flow distance compared to the GA patch in continuous habitat. (4) Anthropogenic barriers (e.g., roads and farmland) reduce effective dispersal distances by 30% in fragmented patches (e.g., SXFP-FB: 37.03 m) compared to continuous forest patches (e.g., SXFP−FA: 66.12 m), likely due to disrupted pollinator movement. These findings underscore the necessity for conservation strategies that address both biotic (pollinator corridors) and abiotic (slope stabilization vegetation) constraints on gene flow. For instance, in the construction of future in situ corridors, specific measures should be tailored to populations with strong SGS (e.g., BMXS, *Sp* = 0.021). Corridors with a width of no less than 50 m ought to be established to link fragmented patches (such as YC and YB patches within the BMXS population), thereby facilitating seed and pollen dispersal. Within these corridors, habitats of native pollinators (e.g., nesting sites of Xylocopa dissimilis, the blue carpenter bee) must be conserved to enhance pollen-mediated gene flow. The effectiveness of the corridors shall be assessed through long-term monitoring (annual surveys) of genetic diversity (using SNP markers) and seedling survival rates, with a 10% increase in H_O_ within five years serving as the success criterion.

### 4.4. The Factors Influencing the Formation of FSGS

The formation of FSGS in *T. sinense* arises from synergistic interactions among restricted gene flow, microhabitat heterogeneity, and historical contingencies. Indirect gene flow estimation revealed that effective gene dispersal distances align more closely with seed-mediated dispersal patterns (Table 6), indicating that seed flow limitations dominate over pollen flow in shaping FSGS. This aligns with findings in *Avena sterilis* and *Liriodendron chinense*, where restricted seed dispersal reinforced kinship clustering [46,47]. Microhabitat variability further amplifies FSGS heterogeneity: divergent soil conditions, moisture gradients, temperature regimes, and biotic interactions among patches create localized selection pressures that filter seedling establishment and survival [48]. For instance, patches with high sapling density (e.g., BMXS−YC: 20 trees/1500 m^2^) exhibited stronger SGS (*Sp* = 0.0107), while fragmented patches with stochastic mortality (e.g., GLGS−GC) showed neutral or disrupted kinship structures. Historical bottlenecks, particularly during the Last Glacial Maximum, exacerbated genetic drift in glacial refugia, reducing effective population sizes and counteracting gene flow diffusion [49]. Fossil evidence from Eocene strata [7] suggests prolonged evolutionary isolation, compounding these effects. Thus, the interplay of contemporary dispersal constraints, microenvironmental filtering, and historical genetic erosion collectively drives the pronounced FSGS observed in *T. sinense* populations.

### 4.5. Implication for Conservation

The pronounced FSGS and low genetic diversity (average H_E_ = 0.101) in *T. sinense* populations, exacerbated by anthropogenic habitat fragmentation, necessitate urgent conservation interventions. In situ management should implement selective thinning in dense subpopulations (e.g., BMXS−YC) to optimize pollen/seed dispersal, particularly critical in populations exhibiting elevated inbreeding (Fis = 0.304). Based on genetic data analyses, we recommend prioritizing the deployment of assisted gene flow for ex situ conservation—specifically from the MGFD population (higher H_E_) to the SXFP population (lowest H_E_ and no saplings). During implementation, caution must be taken to mitigate the risk of outbreeding depression. To this end, sampling intervals between individuals should exceed 115 m in populations with strong SGS (BMXS, GLGS) and 30 m in fragmented populations (MGFD), strategically maximizing genetic diversity while minimizing kinship clustering. Long-term monitoring of age−class dynamics is essential, as sapling mortality (e.g., GLGS) and recruitment bottlenecks in old trees threaten to destabilize genetic architectures. Integrating genomic insights with landscape restoration will be pivotal for sustaining evolutionary potential in this relict species.

## 5. Conclusions

Through the analysis of four sampled populations of *Tetracentron sinense*, this study validated our hypotheses: (1) Microhabitat fragmentation induces dispersal limitation—isolated patches (e.g., GLGS−GB) exhibit a stronger spatial genetic structure (SGS) than contiguous counterparts (e.g., GLGS−GA). (2) Kinship clustering represents a direct outcome of “dispersal restriction” in fragmented habitats, with abiotic limiting factors (e.g., slope gradient) further exacerbating this result. (3) Anthropogenic fragmentation amplifies kinship clustering—patches affected by roads or farmlands display 2–3 fold greater SGS intensity compared to undisturbed patches. These findings affirm the role of both contemporary and historical factors in shaping genetic structure.

Our results indicate that future conservation strategies could encompass: (1) Implementing assisted gene flow between populations with distinct genetic backgrounds (e.g., from MGFD to SXFP), though long-term investigations into outbreeding depression remain necessary. (2) Conducting ex situ sampling at intervals of ≥115 m in populations with high SGS, while the optimal sampling distance requires further validation across additional patches. (3) Undertaking habitat corridor trials in GLGS and BMXS, coupled with monitoring of gene flow dynamics over a 5−10−year timeframe.

## Figures and Tables

**Figure 1 biology-14-01214-f001:**
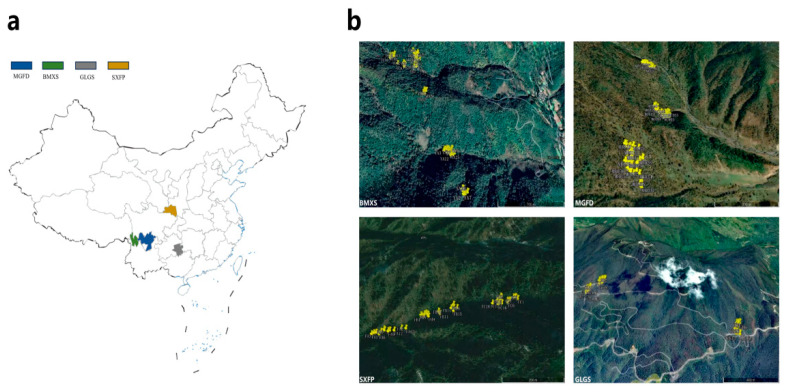
Location, patch, and distribution map of *T. sinense* individuals. (**a**) The sampling sites of the four populations were located in Sichuan Meigu Dafengding (MGFD), Yunnan Baima Snowshan Mountain (BMXS), Guizhou Leigong Mountain (GLGM), and Shanxi Foping (SXFP). (**b**) The distribution of *T. sinense* individuals in three different patches of the four populations. Yellow highlighting is used in the figure to distinguish the sampled individuals from each population.

**Figure 2 biology-14-01214-f002:**
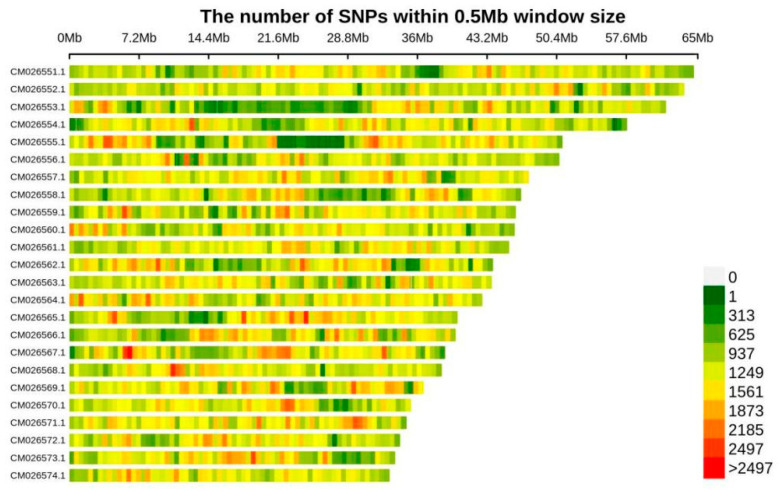
SNP distribution diagram on chromosome. CM026551.1 to CM026574.1 represent the numbering of chromosomes 1 through 24 of *T. sinense*, which has a diploid chromosome number of 48 (2n = 48).

**Figure 3 biology-14-01214-f003:**
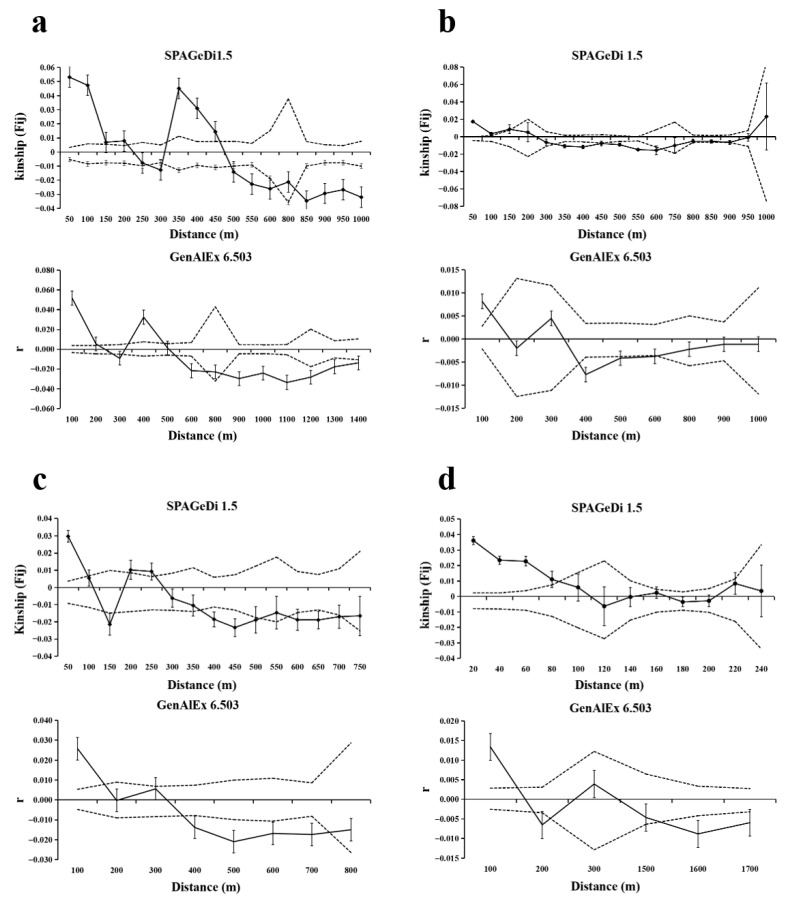
SGS of *T. sinense* in four populations. Solid lines represents genetic coefficients, with virtual lines represents 95% confidence interval and vertical lines represents standard errors. (**a**) SGS in the BMXS population; (**b**) SGS in the MGFD population; (**c**) SGS in the SXFP population; (**d**) SGS in the GLGS population.

**Figure 4 biology-14-01214-f004:**
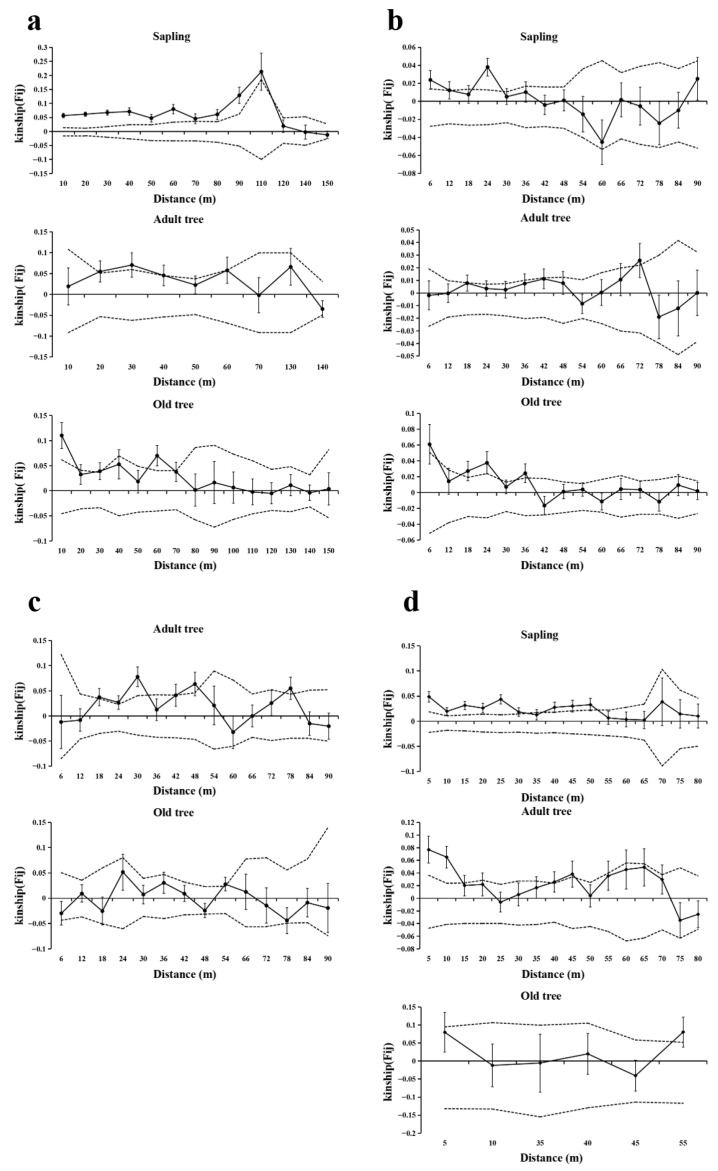
SGS analysis of different age−classes in four populations. Solid lines represents genetic coefficients, with virtual lines represents 95% confidence interval and vertical lines represents standard errors. (**a**) SGS in the BMXS population; (**b**) SGS in the MGFD population; (**c**) SGS in the SXFP population; (**d**) SGS in the GLGS population.

**Figure 5 biology-14-01214-f005:**
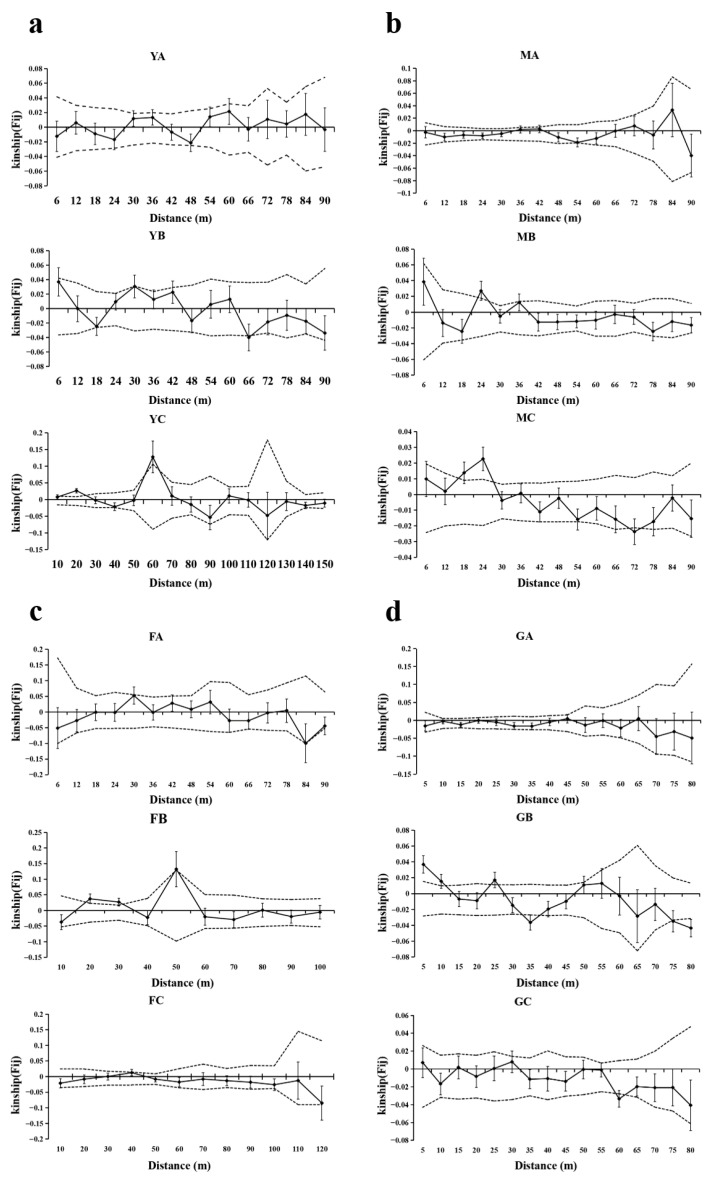
SGS analysis of different patches of four populations. Solid lines represents genetic coefficients, with virtual lines represents 95% confidence interval and vertical lines represents standard errors. (**a**) SGS in the BMXS population; (**b**) SGS in the MGFD population; (**c**) SGS in the SXFP population; (**d**) SGS in the GLGS population.

**Figure 6 biology-14-01214-f006:**
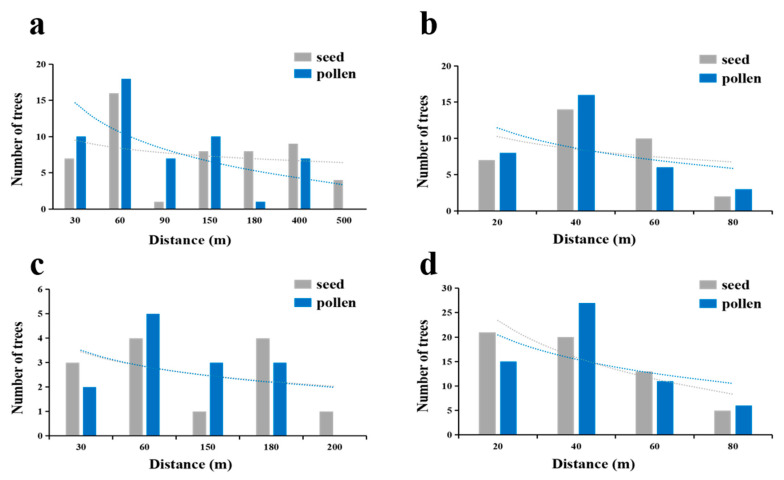
Propagation frequency of seeds and pollen in four populations. The blue line represents pollen, and the gray line represents seed. The dispersal rate of seeds and pollen decreased with the increase in distance between individuals. (**a**) SGS in the BMXS population; (**b**) SGS in the MGFD population; (**c**) SGS in the SXFP population; (**d**) SGS in the GLGS population.

**Figure 7 biology-14-01214-f007:**
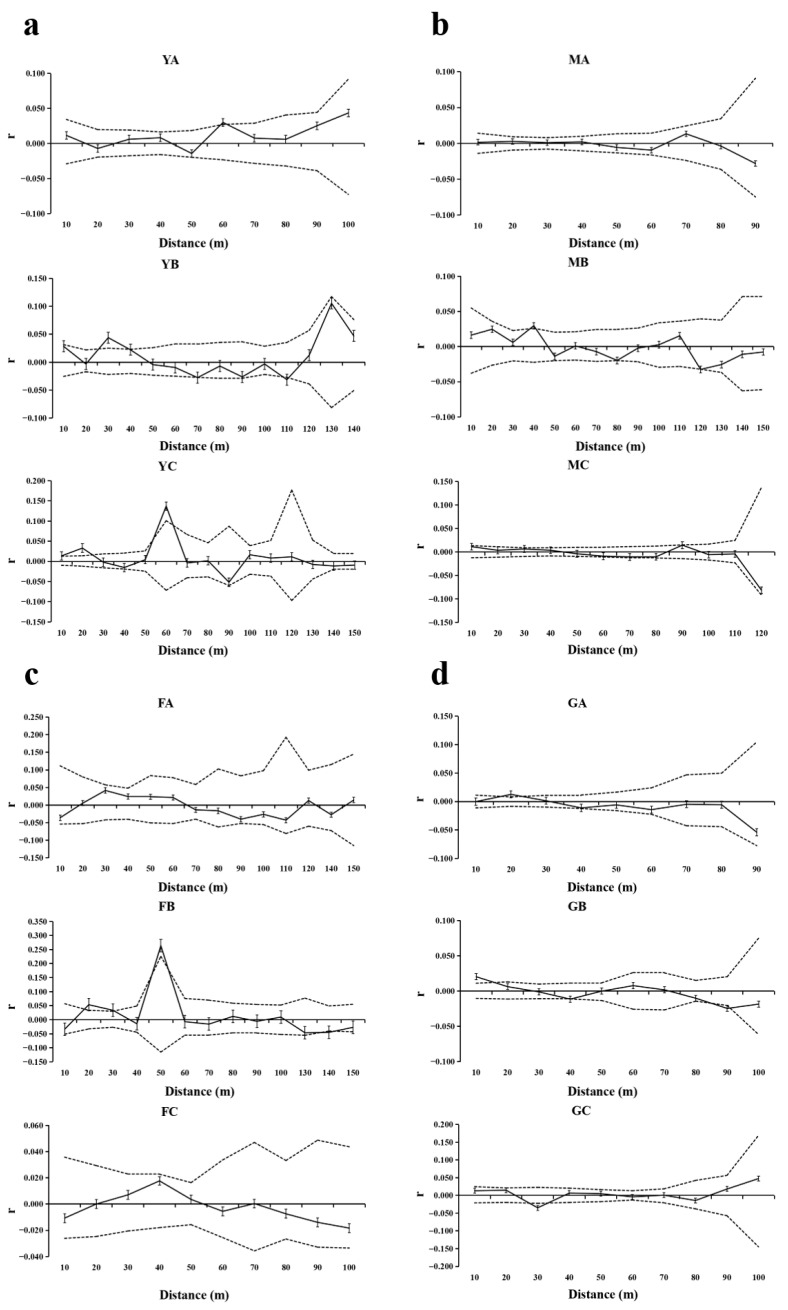
Spatial autocorrelation analysis of different patches of four populations. Solid lines represents genetic coefficients, with virtual lines represents 95% confidence interval and vertical lines represents standard errors. (**a**) SGS in the BMXS population; (**b**) SGS in the MGFD population; (**c**) SGS in the SXFP population; (**d**) SGS in the GLGS population.

**Table 1 biology-14-01214-t001:** Genetic diversity of *T. sinense*.

Pop ID.		Num Indv	Ho	H_E_	Fis
BMXS	YA	9.470	0.020	0.083	0.171
	YB	8.946	0.019	0.083	0.172
	YC	8.870	0.018	0.084	0.178
	means	9.095	0.019	0.083	0.174
GLGS	GA	7.796	0.017	0.111	0.252
	GB	7.602	0.016	0.106	0.241
	GC	7.328	0.024	0.112	0.237
	means	7.575	0.019	0.110	0.243
MGFD	MA	10.644	0.022	0.134	0.314
	MB	8.861	0.022	0.125	0.285
	MC	10.674	0.020	0.132	0.314
	means	10.060	0.021	0.130	0.304
SXFP	FA	4.356	0.022	0.075	0.136
	FB	4.866	0.020	0.078	0.149
	FC	5.269	0.025	0.084	0.155
	means	4.830	0.022	0.079	0.147

Genetic diversity of *T. sinense* at different patch levels. Num Indv is the average number of individuals per locus, H_O_ is the observed heterozygosity, and H_E_ is the unbiased expected heterozygosity. The level of genetic diversity was determined by H_E_. Fis is the inbreeding coefficient, which measures the degree of kinship between the parents of the individual, reflecting the degree of inbreeding.

**Table 2 biology-14-01214-t002:** *t*-test for genetic diversity of *T. sinense*.

Pop ID	Mean Difference (H_O_−H_E_)	*t*-Value	Degrees of Freedom (df)	*p*-Value	Significance
BMXS	−0.064	−96.5	4	<0.001	***
GLGS	−0.091	−28.99	4	<0.001	***
MGFD	−0.109	−38.81	4	<0.001	***
SXFP	−0.057	−18.77	4	<0.001	***

Independent samples *t*-test for genetic diversity of four populations. The “level of significance” of statistical test results is usually determined based on the *p*-value: *p* < 0.05: “Significant” (commonly marked with *); *p* < 0.01: “Highly significant” (commonly marked with **); *p* < 0.001: “Extremely significant” (commonly marked with ***); *p* ≥ 0.05: “Not significant”.

**Table 3 biology-14-01214-t003:** SGS analysis at the population level of *T. sinense*.

Populations	*bF*	F_(1)_	*Sp*
BMXS	−0.0204	0.0531	0.021
MGFD	−0.0075	0.0176	0.0076
SXFP	−0.0133	0.0297	0.013
GLGS	−0.012	0.0362	0.012

SGS analysis of four populations. F_(1)_ is the average value of the genetic relationship between individuals in the first distance level, and *bF* is the linear regression slope of the genetic relationship to the natural logarithm of the distance level. *Sp* quantifies the intensity of SGS.

**Table 4 biology-14-01214-t004:** SGS analysis at the age−class level of *T. sinense*.

Populations	Age-Classes	*bF*	F_(1)_	*Sp*
BMXS	Sapling	−0.0218	0.0566	0.023
	Adult tree	−0.0177	0.0193	0.018
	Old tree	−0.0194	0.1104	0.022
MGFD	Sapling	−0.0101	0.0237	0.010
	Adult tree	−0.0041	−0.0019	0.004
	Old tree	−0.0100	0.0610	0.010
SXFP	Sapling	−	−	−
	Adult tree	0.0134	−0.0120	0.010
	Old tree	0.0052	−0.0298	0.005
GLGS	Sapling	−0.0120	0.0487	0.010
	Adult tree	−0.0119	0.0768	0.010
	Old tree	−0.0175	0.0800	0.020

SGS analysis of different age−classes in four populations. F_(1)_ is the average value of the genetic relationship between individuals in the first distance level, and *bF* is the linear regression slope of the genetic relationship to the natural logarithm of the distance level. *Sp* quantifies the intensity of SGS.

**Table 5 biology-14-01214-t005:** SGS analysis at the patch level of *T. sinense*.

Populations	Patches	*bF*	F_(1)_	*Sp*
BMXS	YA	0.0031	−0.0124	0.0030
	YB	−0.0031	0.0368	0.0032
	YC	0.0107	0.0073	0.0107
MGFD	MA	0.0006	−0.0026	0.0006
	MB	0.0060	0.0385	0.0063
	MC	0.0118	0.0099	0.0119
SXFP	FA	0.0108	−0.0516	0.0103
	FB	0.0184	−0.0371	0.0177
	FC	−0.0021	−0.0211	0.0020
GLGS	GA	0.0008	−0.0156	0.0007
	GB	0.0171	0.0368	0.0177
	GC	0.0066	0.0071	0.0066

SGS analysis of different patches in four populations. F_(1)_ is the average value of the genetic relationship between individuals in the first distance level, and *bF* is the linear regression slope of the genetic relationship to the natural logarithm of the distance level. *Sp* quantifies the size of SGS.

**Table 6 biology-14-01214-t006:** Direct estimation of gene flow in *T. sinense*.

Populations	Patches	Maximum Distance of Seed Dispersal/m	Mean Distance of Seed Dispersal/m	Maximum Distance of Pollen Dispersal/m	Mean Distance of Pollen Dispersal/m	Efficient Distance of Gene Dispersal/m
BMXS	YA	398.57	168.87	386.20	201.89	221.13
	YB	463.52	200.89	133.01	55.05	204.63
	YC	165.44	112.10	167.54	74.11	123.74
	means	−	160.62	−	110.35	183.17
MGFD	MA	61.76	28.97	49.16	27.00	34.70
	MB	−	−	−	−	−
	MC	61.00	35.91	78.94	34.17	43.28
	means	−	32.44	−	30.59	38.99
SXFP	FA	−	−	−	−	−
	FB	187.69	117.78	156.67	101.90	138.07
	FC	82.06	39.68	74.54	39.37	48.47
	means	−	78.73	−	70.64	93.27
GLGS	GA	63.66	28.54	39.53	25.61	33.80
	GB	75.62	22.27	68.78	31.87	31.68
	GC	67.09	37.99	84.06	42.56	48.47
	means	−	29.60	−	33.35	37.98

Direct estimation of gene flow. The average distance of seed, pollen, and gene dispersal in different patches of four populations.

**Table 7 biology-14-01214-t007:** Indirect estimation of gene flow in *T. sinense*.

Populations	Patches	Distance of Seed Dispersal		Distance of Pollen Dispersal		Distance of Gene Dispersal/m
		**Min/m**	**Max/m**	**Min/m**	**Max/m**	
BMXS	YA	10.10	16.14	−	10.10	16.14
	YB	13.24	18.97	−	13.24	18.97
	YC	20.71	29.42	−	20.71	29.42
	means	14.68	21.51	−	14.68	21.51
MGFD	MA	34.83	42.66	−	34.83	42.66
	MB	38.20	46.81	−	38.20	46.81
	MC	36.76	45.05	−	36.76	45.05
	means	36.60	44.84	−	36.60	44.84
SXFP	FA	53.95	66.12	−	53.95	66.12
	FB	30.22	37.03	−	30.22	37.03
	FC	43.95	53.86	−	43.95	53.86
	means	42.71	52.34	−	42.71	52.34
GLGS	GA	25.30	31.00	−	25.30	31.00
	GB	23.20	28.43	−	23.20	28.43
	GC	17.63	23.00	−	17.63	23.00
	means	22.04	27.48	−	22.04	27.48

Indirect estimation of gene flow. The distance of pollen and seed dispersal in different patches of four populations.

## Data Availability

The original contributions presented in this study are included in the article/Appendix A. Further inquiries can be directed to the corresponding author.

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
