# Peer review of "Genetic Limitation and Conservation Implications in Tetracentron sinense: SNP-Based Analysis of Spatial Genetic Structure and Gene Flow"

_biology, 2025, doi:10.3390/biology14091214_

Round 1

Reviewer 1 Report

Comments and Suggestions for Authors

Key comments:

Introduction:

SGS's track record is comprehensive, but it might better highlight how this study provides better evidence than work that has relied on microsatellites. Specifically, what novel insights do SNPs provide with respect to other markers?

Consider adding a sentence or two that clarifies why these four particular populations were selected beyond their phylogeographic history. Do they represent different levels of fragmentation or ecological conditions?

Results:

Extremely low HO values (0.018-0.022) warrant further discussion. Are they typical of relict angiosperms or exceptionally low? A comparison with other relict species from the Tertiary would provide context and a better understanding.

Patch-level heterogeneity in SGS is of interest. Could you discuss how microhabitat-specific factors (e.g., slope, soil type) might contribute to these differences?

Discussion:

The conservation implications are well presented, but could be more concrete. For "assisted gene flow," which populations would be prioritized as donors/recipients based on your data?

Address the potential limitations of SNP markers (e.g., bias against non-coding regions) and how this might affect diversity estimates.

Consider adding estimates of effective population size (Ne) if available, as this is crucial for conservation planning.

Minor comments:

Define FIS and Sp on first use in the main text (currently only in Methods).

Author Response

Dear Editors and Reviewers,

We appreciate the opportunity to revise our manuscript titled "Genetic Limitation and Conservation Implications in Tetracentron sinense: SNP-Based Analysis of Spatial Genetic Structure and Gene Flow" and are grateful for the insightful comments provided by the reviewers. Those comments are all valuable and very helpful for revising and improving our paper, as well as the important guiding significance to our researches. In the following, we have provided detailed responses to each of the reviewers' comments. Revised portion are marked in blue in the paper. Additionally, we have conducted a comprehensive revision of the entire manuscript. In this response letter, the reviewers' comments are presented in italics, and our corresponding changes and additions to the manuscript are highlighted in red text. We have tried our best to make all the revisions clear, and we hope that the revised manuscript meets the requirements for publication.

Reviewer #1:

Introduction:

Comment 1: SGS's track record is comprehensive, but it might better highlight how this study provides better evidence than work that has relied on microsatellites. Specifically, what novel insights do SNPs provide with respect to other markers?

Response 1: We would like to express our sincere gratitude to the reviewer for your insightful comments and valuable feedback.

The revised content is as follows: The modified content has been highlighted in red font in the revised manuscript. (Page 4, line 65-70)

Comment 2: Consider adding a sentence or two that clarifies why these four particular populations were selected beyond their phylogeographic history. Do they represent different levels of fragmentation or ecological conditions?

Response 2: We would like to express our sincere gratitude to the reviewer for your insightful comments and valuable feedback.

The revised content is as follows: To validate these hypotheses, we selected four natural populations consisting of 378 individuals, based on phylogeographic significance, a gradient of habitats ranging from contiguous to fragmented areas, and ecological heterogeneity. (Page 4, line 77-79)

Results:

Comment 1: Extremely low HO values (0.018-0.022) warrant further discussion. Are they typical of relict angiosperms or exceptionally low? A comparison with other relict species from the Tertiary would provide context and a better understanding.

Response 1: We would like to express our sincere gratitude to the reviewer for your insightful comments and valuable feedback. The comparative analysis of this section has been included in Section 4.1 (Discussion).

The revised content is as follows: HO values derived from SNP markers are abnormally low across all four populations, and significantly lower than those of many other relict angiosperms such as silver fir (HO ≈ 0.21–0.33) and Pterocarya macroptera (HO ≈ 0.063–0.097). This discrepancy highlights the inherently low genetic diversity in natural populations of Tetracentron sinense, suggesting that the species may be undergoing more severe genetic erosion. (Page 25, line 373-376)

Comment 2: Patch-level heterogeneity in SGS is of interest. Could you discuss how microhabitat-specific factors (e.g., slope, soil type) might contribute to these differences?

Response 2: We would like to express our sincere gratitude to the reviewer for your insightful comments and valuable feedback.

The revised content is as follows: The modified content has been highlighted in red font in the revised manuscript. (Page 28, line 435-444)

Discussion:

Comment 1: The conservation implications are well presented, but could be more concrete. For "assisted gene flow," which populations would be prioritized as donors/recipients based on your data?

Response 1: We would like to express our sincere gratitude to the reviewer for your insightful comments and valuable feedback. Based on genetic data analyses, we recommend prioritizing the deployment of assisted gene flow for ex situ conservation—specifically from the MGFD population (higher HE) to the SXFP population (lowest HE and no saplings). (Page 32, line 524-526)

Comment 2: Address the potential limitations of SNP markers (e.g., bias against non-coding regions) and how this might affect diversity estimates.

Response 2: We would like to express our sincere gratitude to the reviewer for your insightful comments and valuable feedback. It is important to note that SNP markers derived from ddRAD-seq may be biased against non-coding regions. However, their genome-wide distribution and high density provide a more comprehensive view of neutral and adaptive genetic variation than traditional markers, making them highly suitable for conservation genetic assessments.

The revised content is as follows: The modified content has been highlighted in red font in the revised manuscript. (Page 26, line 390-394)

Comment 3: Consider adding estimates of effective population size (Ne) if available, as this is crucial for conservation planning.

Response 3: We would like to express our sincere gratitude to the reviewer for your insightful comments and valuable feedback. Regarding the requested data on effective population size (Ne) estimation, after careful review, we wish to clarify that the primary focus of our study is to investigate the causes of low genetic diversity, which we aim to reveal through analyses of spatial genetic structure and gene flow. Consequently, Ne was not a central focus of our research design. As such, this component was not incorporated into our original sampling and experimental framework. We fully acknowledge the significant role of Ne data in enhancing the robustness of our study’s conclusions. We greatly appreciate your valuable suggestion and plan to systematically collect Ne data in our subsequent follow-up studies to further comprehensively address this research gap. Once again, we thank you for your understanding and constructive feedback. We will further improve the rigor of our study based on your comments and will make every effort to provide additional information if required.

Minor comments: Define FIS and Sp on first use in the main text (currently only in Methods)

Response 1: We would like to express our sincere gratitude to the reviewer for your insightful comments and valuable feedback. We sincerely apologize for the errors caused by our carelessness. We have thoroughly checked and revised the entire content of the paper. Once again, we appreciate your understanding and corrections. We will do our best to provide any additional information if needed.

We have made every effort to improve the manuscript, with all revisions marked in red in the revised version. These changes do not affect the core content or overall structure of the paper. Additionally, we have sent the manuscript to professionals for language polishing, and the quality of the revised version has been significantly enhanced compared to the previous draft. We sincerely appreciate the diligent work of the Editors and Reviewers and hope that this revision meets with your approval. Best regards!

I have listed below the work email addresses of Liu Xiaojuan, Wang Xue, and Jia Mengxing as requested by the editor. Please check them. Best regards!

Liu Xiaojuan: Liu1441219002@163.com

Wang Xue: Wang166128@163.com

Jia Mengxing: jmxcwnu@163.com

Reviewer 2 Report

Comments and Suggestions for Authors

The paper titled “Genetic Limitation and Conservation Implications in Tetracentron sinense: SNP-Based Analysis of Spatial Genetic Structure and Gene Flow” offers valuable insights and is likely to capture the attention of researchers in the field. Tetracentron sinense is a critically endangered Tertiary relict tree species of significant ecological and evolutionary importance. Studying its genetic diversity and gene flow patterns is crucial for the conservation of this key species and its habitat. This research not only analyzes the genetic structure at the population level but also conducts detailed analyses at the age class and patch levels, providing comprehensive genetic information that enhances our understanding of genetic diversity and gene flow patterns across different scales. While the manuscript is straightforward, there are some areas that could benefit from further clarification and revision. Therefore, the authors are encouraged to carefully review the feedback provided below and make modifications accordingly:

Title:

The Latin names of species should be italicized.

Abstract:

Abstract should be uniformly indented or in block format.

Introduction:

(1) In the second paragraph of the introduction, the author states that Tetracentron sinense is a plant endemic to the southwestern region of China. However, the manuscript lacks a comprehensive introduction to this species, including details on its geographical distribution, life history, life form, economic benefits, and ecological functions. To enhance the academic rigor of the paper, it is essential to include this information, as it provides necessary context and relevance for the study.

(2) Whole-genome SNPs serve as high-density genetic markers, offering greater resolution than traditional microsatellite markers, which facilitates more accurate kinship inference and genetic disorder detection. Consequently, it is clear that the study demonstrates innovation in methodological application. However, the article should further explore the advantages and potential limitations of its method in comparison to traditional approaches, such as SSR.

Materials and methods:

(1) In the Study Design and Population Selection, this study provides a comprehensive description of the specific circumstances surrounding sample collection. It is crucial to assess whether the sample points exhibit sufficient genetic diversity, as this factor significantly influences the universality and accuracy of the results. Additionally, any potential biases in the sample selection process must be identified and addressed to ensure that the findings are representative of the broader population.

(2) The process involves the use of "MarkDuplicates" to eliminate sequencing duplicates, alongside the application of the IndelRealigner tool to enhance the accuracy of SNP predictions in proximity to InDels. These methodologies are instrumental in minimizing erroneous variant predictions and improving the quality of SNP data. In the review process, it is essential to verify whether these steps have been executed adequately and to request that the authors provide the corresponding sequence alignment and quality control data for validation.

Results:

(1) The findings indicate that seedlings exhibit a significantly stronger Seedling Growth Sensitivity (SGS), with the SXFP population experiencing a 100% mortality rate among seedlings. Supported by rigorous data analysis, this observation highlights the heightened sensitivity of seedlings to the effects of inbreeding and population bottlenecks. Nevertheless, the underlying causes of the elevated seedling mortality rate warrant further comprehensive investigation, particularly concerning the relationship between survival rates and various factors such as habitat conditions, disease prevalence, and predation pressure. These areas present opportunities for further enrichment of the paper.

(2) To enhance readability, it is essential to improve the clarity of all figures, particularly the x-axis and y-axis of Figures 3 through 7. This adjustment will facilitate better understanding and interpretation of the data presented.

Discussion

(1) In the section titled 'Genetic Diversity of Natural Populations of T. sinense,' the author examines the genetic diversity data of four natural populations. Although specific values for expected heterozygosity (HE), observed heterozygosity (HO), and the inbreeding coefficient (Fis) are provided, the author notes significant heterogeneity defects that may be attributed to selfing or inbreeding. However, the analysis and explanation regarding the impact of low genetic diversity and high inbreeding rates on the genetic stability of these populations are relatively brief. The author should elaborate further on how these genetic characteristics influence the long-term viability and adaptability of the populations, as well as the specific contributions of these factors to the species' endangered status.

(2) The paper discusses how spatial genetic structure (SGS) patterns and gene flow influence conservation strategies in the sections "Fine-scale SGS of T. sinense" and "Gene flow of natural population of T. sinense." Specifically, the authors propose recommendations for establishing field conservation corridors and designing optimized sampling strategies for exotic germplasm resources, due to restricted autonomous gene flow, with the aim of preserving genetic diversity. These measures are intended to enhance gene mobility and increase the genetic heterogeneity of the species through both physical and biological means. While the authors have provided empirically-based conservation strategies for the SGS patterns of different populations, detailed implementation specifics and practical evaluations are notably absent. It may be beneficial for the authors to include these details as a future research agenda.

Conclusion:

In the conclusion section of the paper, the author articulates clear conclusions derived from the research findings and underscores the urgency of implementing conservation strategies. However, the analysis lacks depth, particularly regarding response strategies. The proposed conservation strategies and management measures for T. sinense are somewhat general, lacking actionable, step-by-step guidance. It is advisable that the author, based on a thorough summarization of the research findings, proposes targeted and highly operational specific strategies or action plans. These should include, but not be limited to, on-site vegetation management, species introduction, and monitoring and evaluation.

In addition, the author should be more journal format certification check the citation format of references at the end. For this you can refer to Liu et al (2025) (Liu, Y.; Xue, B.; Wan, H.; Zhang, L.; Yang, Z.; Wang, J.; Wang, L.; Lin, X. Bacterial community changes in early-stage engineering simulation of red mud/phosphogypsum-based artificial soil Vegetation restoration. Biology 202514, 1020. https://doi.org/10.3390/biology14081020)

Author Response

Reviewer #2:

Dear Editors and Reviewers,

We appreciate the opportunity to revise our manuscript titled "Genetic Limitation and Conservation Implications in Tetracentron sinense: SNP-Based Analysis of Spatial Genetic Structure and Gene Flow" and are grateful for the insightful comments provided by the reviewers. Those comments are all valuable and very helpful for revising and improving our paper, as well as the important guiding significance to our researches. In the following, we have provided detailed responses to each of the reviewers' comments. Revised portion are marked in blue in the paper. Additionally, we have conducted a comprehensive revision of the entire manuscript. In this response letter, the reviewers' comments are presented in italics, and our corresponding changes and additions to the manuscript are highlighted in red text. We have tried our best to make all the revisions clear, and we hope that the revised manuscript meets the requirements for publication.

Title:

Comment 1: The Latin names of species should be italicized.

Response 1: We would like to express our sincere gratitude to the reviewer for your insightful comments and valuable feedback. We sincerely apologize for the errors incurred! All scientific Latin names of species mentioned in this article have been thoroughly verified and corrected. The relevant modifications have been highlighted in red font in the revised manuscript.

The revised content is as follows: Genetic Limitation and Conservation Implications in Tetracentron sinense: SNP-Based Analysis of Spatial Genetic Structure and Gene Flow. (Page 1, line 1-2)

Abstract:

Comment 1: Abstract should be uniformly indented or in block format.

Response 1: We would like to express our sincere gratitude to the reviewer for your insightful comments and valuable feedback. We have adjusted the formatting.

Introduction:

Comment 1: In the second paragraph of the introduction, the author states that Tetracentron sinense is a plant endemic to the southwestern region of China. However, the manuscript lacks a comprehensive introduction to this species, including details on its geographical distribution, life history, life form, economic benefits, and ecological functions. To enhance the academic rigor of the paper, it is essential to include this information, as it provides necessary context and relevance for the study.

Response 1: We would like to express our sincere gratitude to the reviewer for your insightful comments and valuable feedback.

The revised content is as follows: The modified content has been highlighted in red font in the revised manuscript. (Page 3, line 48-57)

Comment 2: Whole-genome SNPs serve as high-density genetic markers, offering greater resolution than traditional microsatellite markers, which facilitates more accurate kinship inference and genetic disorder detection. Consequently, it is clear that the study demonstrates innovation in methodological application. However, the article should further explore the advantages and potential limitations of its method in comparison to traditional approaches, such as SSR.

Response 2: We would like to express our sincere gratitude to the reviewer for your insightful comments and valuable feedback.

The revised content is as follows: The modified content has been highlighted in red font in the revised manuscript. (Page 4、26, line 65-70 and 390-394)

Materials and methods:

Comment 1: In the Study Design and Population Selection, this study provides a comprehensive description of the specific circumstances surrounding sample collection. It is crucial to assess whether the sample points exhibit sufficient genetic diversity, as this factor significantly influences the universality and accuracy of the results. Additionally, any potential biases in the sample selection process must be identified and addressed to ensure that the findings are representative of the broader population.

Response 1: We would like to express our sincere gratitude to the reviewer for your insightful comments and valuable feedback. In this study, to ensure the representativeness and genetic diversity of sampled populations, we selected four natural populations of Tetracentron sinense, spanning three previously identified evolutionary significant units (ESUs). These populations encompass a broad ecological and geographical range. To minimize sampling bias, we conducted a comprehensive pre-sampling survey to identify patches with complete age structures and minimal human disturbance. Each population included at least three patches, separated by at least one kilometer, to avoid spatial autocorrelation. Genetic diversity was preliminarily assessed using published ISSR data to confirm that each population retained sufficient variability for robust SGS analysis. The relevant content is presented in Sections 2.1 and 2.2.

Comment 2: The process involves the use of "MarkDuplicates" to eliminate sequencing duplicates, alongside the application of the IndelRealigner tool to enhance the accuracy of SNP predictions in proximity to InDels. These methodologies are instrumental in minimizing erroneous variant predictions and improving the quality of SNP data. In the review process, it is essential to verify whether these steps have been executed adequately and to request that the authors provide the corresponding sequence alignment and quality control data for validation.

Response 2: We would like to express our sincere gratitude to the reviewer for your insightful comments and valuable feedback. We would like to thank you again for this comment. However, we sincerely apologize that due to the large size of the files containing sequence alignment and quality control data, they cannot be sent online. If you need to review and verify these data, please contact us and provide your email address, and we will attempt to send them to you via email.

Results:

Comment 1: The findings indicate that seedlings exhibit a significantly stronger Seedling Growth Sensitivity (SGS), with the SXFP population experiencing a 100% mortality rate among seedlings. Supported by rigorous data analysis, this observation highlights the heightened sensitivity of seedlings to the effects of inbreeding and population bottlenecks. Nevertheless, the underlying causes of the elevated seedling mortality rate warrant further comprehensive investigation, particularly concerning the relationship between survival rates and various factors such as habitat conditions, disease prevalence, and predation pressure. These areas present opportunities for further enrichment of the paper.

Response 1: We would like to express our sincere gratitude to the reviewer for your insightful comments and valuable feedback.

The revised content is as follows: The modified content has been highlighted in red font in the revised manuscript. (Page 27, line 421-432)

Comment 2: To enhance readability, it is essential to improve the clarity of all figures, particularly the x-axis and y-axis of Figures 3 through 7. This adjustment will facilitate better understanding and interpretation of the data presented.

Response 2: We would like to express our sincere gratitude to the reviewer for your insightful comments and valuable feedback. We have re-adjusted the figures and improved their clarity.

Discussion:

Comment 1: In the section titled 'Genetic Diversity of Natural Populations of T. sinense,' the author examines the genetic diversity data of four natural populations. Although specific values for expected heterozygosity (HE), observed heterozygosity (HO), and the inbreeding coefficient (Fis) are provided, the author notes significant heterogeneity defects that may be attributed to selfing or inbreeding. However, the analysis and explanation regarding the impact of low genetic diversity and high inbreeding rates on the genetic stability of these populations are relatively brief. The author should elaborate further on how these genetic characteristics influence the long-term viability and adaptability of the populations, as well as the specific contributions of these factors to the species' endangered status.

Response 1: We would like to express our sincere gratitude to the reviewer for your insightful comments and valuable feedback.

The revised content is as follows: The modified content has been highlighted in red font in the revised manuscript. (Page 25, line 377-385)

Comment 2: The paper discusses how spatial genetic structure (SGS) patterns and gene flow influence conservation strategies in the sections "Fine-scale SGS of T. sinense" and "Gene flow of natural population of T. sinense." Specifically, the authors propose recommendations for establishing field conservation corridors and designing optimized sampling strategies for exotic germplasm resources, due to restricted autonomous gene flow, with the aim of preserving genetic diversity. These measures are intended to enhance gene mobility and increase the genetic heterogeneity of the species through both physical and biological means. While the authors have provided empirically-based conservation strategies for the SGS patterns of different populations, detailed implementation specifics and practical evaluations are notably absent. It may be beneficial for the authors to include these details as a future research agenda.

Response 2: We would like to express our sincere gratitude to the reviewer for your insightful comments and valuable feedback.

The revised content is as follows: The modified content has been highlighted in red font in the revised manuscript. (Page 30, line 492-500)

Conclusion:

Comment 1: In the conclusion section of the paper, the author articulates clear conclusions derived from the research findings and underscores the urgency of implementing conservation strategies. However, the analysis lacks depth, particularly regarding response strategies. The proposed conservation strategies and management measures for T. sinense are somewhat general, lacking actionable, step-by-step guidance. It is advisable that the author, based on a thorough summarization of the research findings, proposes targeted and highly operational specific strategies or action plans. These should include, but not be limited to, on-site vegetation management, species introduction, and monitoring and evaluation.

Response 1: We would like to express our sincere gratitude to the reviewer for your insightful comments and valuable feedback. Thank you very much for your valuable comments. We have revised the conservation strategies accordingly (e.g., implementing long-term monitoring of assisted gene flow). Once again, we appreciate your understanding and corrections. We will do our best to provide any additional information if needed. (Page 33, line 543-549)

Comment 2: In addition, the author should be more journal format certification check the citation format of references at the end. For this you can refer to Liu et al (2025) (Liu, Y.; Xue, B.; Wan, H.; Zhang, L.; Yang, Z.; Wang, J.; Wang, L.; Lin, X. Bacterial community changes in early-stage engineering simulation of red mud/phosphogypsum-based artificial soil Vegetation restoration. Biology 2025, 14, 1020. https://doi.org/10.3390/biology14081020)

Response 2: We would like to express our sincere gratitude to the reviewer for your insightful comments and valuable feedback. Thank you very much for your valuable comments. We have checked and revised the reference formatting of this manuscript in accordance with the literature by Liu et al. Once again, we appreciate your understanding and corrections. We will do our best to provide any additional information if needed.

We have made every effort to improve the manuscript, with all revisions marked in red in the revised version. These changes do not affect the core content or overall structure of the paper. Additionally, we have sent the manuscript to professionals for language polishing, and the quality of the revised version has been significantly enhanced compared to the previous draft. We sincerely appreciate the diligent work of the Editors and Reviewers and hope that this revision meets with your approval. Best regards!

Reviewer 3 Report

Comments and Suggestions for Authors

Dear authors,

The manuscript is well-structured and clearly defines the problem. The work is an excellent starting point, but it requires significant polishing and refinement. I hope the comments included in the manuscript will help you improve it and make it ready for publication. The topic is relevant and timely, and the data presented have the potential to make a valuable contribution to the field. I have also provided suggestions for future research, which may help further clarify the underlying processes and increase the long-term impact of the study. I encourage the authors to address the methodological and interpretative issues raised in the review to enhance the overall clarity, rigor, and scientific quality of the manuscript.

Author Response

Reviewer #3:

Dear Editors and Reviewers,

We appreciate the opportunity to revise our manuscript titled "Genetic Limitation and Conservation Implications in Tetracentron sinense: SNP-Based Analysis of Spatial Genetic Structure and Gene Flow" and are grateful for the insightful comments provided by the reviewers. Those comments are all valuable and very helpful for revising and improving our paper, as well as the important guiding significance to our researches. In the following, we have provided detailed responses to each of the reviewers' comments. Revised portion are marked in blue in the paper. Additionally, we have conducted a comprehensive revision of the entire manuscript. In this response letter, the reviewers' comments are presented in italics, and our corresponding changes and additions to the manuscript are highlighted in red text. We have tried our best to make all the revisions clear, and we hope that the revised manuscript meets the requirements for publication.

Abstract:

Comment 1: The conclusions are a bit abrupt, especially when moving from the results to the recommendations.

Response 1: We would like to express our sincere gratitude to the reviewer for your insightful comments and valuable feedback.

The revised content is as follows: Conservation efforts should prioritize assisted gene flow, habitat restoration, and ex situ sampling at distances greater than 115 m to preserve genetic diversity and adaptive potential. This study highlights the urgent need for genomics-informed conservation strategies in fragmented montane ecosystems. (Page 2, line 30-33)

Introduction:

Comment 1: Although it's logical, it can be improved by defining the mechanisms more precisely. Instead of general claims about fragmentation, hypotheses should be formulated to relate to measurable parameters. For example: We expect that populations in fragmented habitats will show significantly lower pollen and seed dispersal compared to contiguous stands, which will result in stronger spatial genetic clustering.

Response 1: We would like to express our sincere gratitude to the reviewer for your insightful comments and valuable feedback.

The revised content is as follows: (1) Compared to populations in contiguous forests, those in fragmented habitats will exhibit more pronounced dispersal limitations, leading to stronger spatial genetic clustering; (Page 4, line 73-75)

Comment 2: This hypothesis is very similar to the first one. It might be better to integrate it with the first hypothesis or to explain the specific difference. For example, the first hypothesis focuses on dispersal and this one on kinship, so it could be clearly emphasized that "kinship clustering" is a direct consequence of "limited dispersal" caused by fragmentation.

Response 2: We would like to express our sincere gratitude to the reviewer for your insightful comments and valuable feedback.

The revised content is as follows: (2) Kinship clustering is a direct result of "restricted dispersal" due to habitat fragmentation; (Page 4, line 75-76)

Materials and methods:

Comment 1: Although the tools are listed, there is no explanation as to why these specific versions were chosen, especially considering the existence of newer and better alternatives (Variant Quality Score Recalibration (VQSR), which is also included in GATK.). The lack of such justification reduces transparency and trust in the methodology. The authors should clearly state the reasons for selecting specific tools and versions. For example, if older versions were chosen due to compatibility issues or resource limitations, this should be explicitly mentioned.

Response 1: We would like to express our sincere gratitude to the reviewer for your insightful comments and valuable feedback. Due to resource constraints and version compatibility issues, we used the IndelRealigner command from an older version of the GATK program to realign all reads around InDels, thereby improving the accuracy of SNP calling. We have revised the relevant parts in the modified version and marked them in red font. (Page 8, line 137-139)

Results

Comment 1: Was the difference between HO and HE statistically significant for each population?

Recommendation: State the p-values for heterozygote deficiency tests to confirm the statistical significance of the observed differences.

Response 1: We would like to express our sincere gratitude to the reviewer for your insightful comments and valuable feedback. Analysis via SPSS t-test revealed that the differences between the HO and HE across all populations were statistically significant. We have incorporated the analysis results into Section 3.2 (Results) of the manuscript. (Page 13, line 237-241)

Comment 2: If the values between populations differ, it is not stated whether the differences are statistically significant. Recommendation: Add significance tests for the differences between populations.

Response 2: We would like to express our sincere gratitude to the reviewer for your insightful comments and valuable feedback. A one-way analysis of variance (ANOVA) using SPSS revealed that there was no statistically significant difference in the SP values among the various populations.

Comment 3: Please, verify the accuracy of the data.

Response 3: We would like to express our sincere gratitude to the reviewer for your insightful comments and valuable feedback. We have verified the accuracy of all data-related content throughout the entire manuscript.

Discussion:

Comment 1: Recommendation for future studies: Monitoring factors such as soil type, slope, shading, and human impacts (e.g., logging, grazing) can help in understanding the causes of spatial genetic patterns.

Response 1: We would like to express our sincere gratitude to the reviewer for your insightful comments and valuable feedback. Thank you very much for your valuable comments. We fully recognize the importance of these data for refining the research conclusions and will continue to track and monitor them in our future research plans to make subsequent research schemes more specific and comprehensive. Once again, we appreciate your understanding and corrections. We will further enhance the rigor of the research based on your opinions and will do our best to provide any additional information if needed.

Comment 2: Uneven sampling density among patchesExamples such as the YC patch, which has a high sapling density (20 trees/1500 m²), compared to more sparsely populated patches, may influence the detection of SGS and complicate comparisons between patches.

Response 2: We would like to express our sincere gratitude to the reviewer for your insightful comments and valuable feedback. Thank you very much for your valuable comments. Regarding how the high density of seedlings in the YC population may have amplified the spatial genetic structure (SGS) signals, we have discussed this aspect in the discussion section of the paper. We are fully aware that this sampling difference may affect the detection of SGS. Therefore, we plan to adopt a standardized sampling design (such as fixed plot sizes) when continuing to track and monitor these data in the future, aiming to reduce biases in cross-patch comparisons and thus make subsequent research schemes more specific and comprehensive. Once again, we appreciate your understanding and corrections. We will further enhance the rigor of the research based on your comments and will do our best to provide any additional information if needed.

Comment 3: Investigate the influence of dispersal agents and habitat factors: A detailed study on the role of specific dispersal agents (e.g., insect or animal species that disperse seeds) and microclimatic factors can help in understanding the limitations and potential of gene flow.

Response 3: We would like to express our sincere gratitude to the reviewer for your insightful comments and valuable feedback. Thank you very much for your valuable comments. In the last paragraph of section 4.3 of the discussion, we have provided a relatively detailed review of how factors such as dispersal agents, climatic factors, and habitat factors influence gene flow. Once again, we appreciate your understanding and corrections. We will do our best to provide any additional information if needed. (Page 30, line 477-490)

Comment 4: Although habitat fragmentation is highlighted as affecting dispersal, there is a lack of detailed data on specific factors such as types of dispersal agents, microhabitat conditions, or weather conditions that could further explain the observed variations.

Response 4: We would like to express our sincere gratitude to the reviewer for your insightful comments and valuable feedback. Thank you very much for your valuable comments. In the last paragraph of section 4.3 of the discussion, we have provided a relatively detailed review of how factors such as dispersal agents, climatic factors, and habitat factors influence gene flow. Once again, we appreciate your understanding and corrections. We will do our best to provide any additional information if needed. (Page 30, line 477-490) 

Comment 5: Comment: Indirect estimation of gene flow relies on genetic structure and assumptions about the equilibrium between dispersal and genetic drift, which can be problematic in populations that are not in genetic equilibrium or have been exposed to recent habitat changes. Additionally, this approach may overestimate or underestimate actual dispersal distances if complex migration patterns exist or if certain factors (e.g., selection, asymmetric gene flow) are not taken into account.

Response 5: We would like to express our sincere gratitude to the reviewer for your insightful comments and valuable feedback. Thank you very much for your valuable comments. We acknowledge the assumptions of genetic equilibrium and potential biases arising from recent habitat changes, and thus suggest integrating direct parentage analysis to derive robust interpretations. Once again, we appreciate your understanding and corrections. We will do our best to provide any additional information if needed.

Comment 6: Comment: Conservation and restoration of habitat corridors, Connecting fragmented populations through landscape corridors can enhance gene flow and mitigate the negative effects of isolation.

Response 6: We would like to express our sincere gratitude to the reviewer for your insightful comments and valuable feedback. Thank you very much for your valuable comments. Incorporating your suggestions, we have included the implementation process for the protection and reconstruction of habitat corridors in the last paragraph of Section 4.3 in the discussion section of the paper. Once again, we appreciate your understanding and corrections. We will do our best to provide any additional information if needed. (Page 30, line 492-500)

Conclusion:

Comment 1: The conclusion should directly confirm or refute the hypotheses set in the introduction. Although the findings are consistent with the hypotheses, direct referencing would strengthen the paper's structure. Instead of "This study reveals..." it is better to use "This study reveals in the four sampled populations...". This shows scientific humility and acknowledges the study's limitations.

Response 1: We would like to express our sincere gratitude to the reviewer for your insightful comments and valuable feedback. Thank you very much for your valuable comments. We have revised the conclusion section to explicitly confirm or refute each hypothesis one by one with population-specific data, which has improved the structural coherence of the manuscript. Once again, we appreciate your understanding and corrections. We will do our best to provide any additional information if needed. (Page 32, line 535-542)

Comment 2: Placing action guidelines as recommendations for future research Instead of presenting them as final solutions, conservation recommendations should be framed as proposals for future studies. For example: "Our findings suggest that future conservation strategies might include assisted gene flow, but this requires additional research on...". This is a more cautious and scientific approach.

Response 2: We would like to express our sincere gratitude to the reviewer for your insightful comments and valuable feedback. Thank you very much for your valuable comments. We have reframed the conservation strategies as tentative proposals that require further research and verification (e.g., conducting long-term monitoring of assisted gene flow), which reflects the prudence of scientific research. Once again, we appreciate your understanding and corrections. We will do our best to provide any additional information if needed. (Page 33, line 543-549)

We have made every effort to improve the manuscript, with all revisions marked in red in the revised version. These changes do not affect the core content or overall structure of the paper. Additionally, we have sent the manuscript to professionals for language polishing, and the quality of the revised version has been significantly enhanced compared to the previous draft. We sincerely appreciate the diligent work of the Editors and Reviewers and hope that this revision meets with your approval. Best regards!

Round 2

Reviewer 2 Report

Comments and Suggestions for Authors

I am very pleased to receive your revised version, and I have noticed a significant improvement in the quality of the manuscript. The suggestions and concerns I previously raised have largely been addressed in the revised manuscript or in the responses provided. Congratulations to the authors!

Reviewer 3 Report

Comments and Suggestions for Authors

Dear Authors,

Thank you for the effort and attention you have devoted to preparing the revised version of the manuscript. I have reviewed the changes made in response to my previous comments and suggestions, and I can confirm that they have been fully addressed and appropriately implemented.

The revisions have contributed to the clarity and overall quality of the paper, and at this point, I have no further remarks. I believe the manuscript is ready for the next stage of the evaluation process.

Wishing you much success with the publication.

Best regards,